# Thermoeconomic Analysis of Subcritical and Supercritical Isobutane Cycles for Geothermal Power Generation

Andrea Arbula Blecich [1] and Paolo Blecich [2,*]

1  Faculty of Economics and Business, University of Rijeka, 51000 Rijeka, Croatia; aarbula@efri.hr
2  Faculty of Engineering, University of Rijeka, 51000 Rijeka, Croatia
*  Correspondence: paolo.blecich@riteh.hr

**Abstract:** This article presents a novel and comprehensive approach for the thermoeconomic evaluation of subcritical and supercritical isobutane cycles for geothermal temperatures of $T_{geo}$ = 100–200 °C. The isobutane cycles are optimized with respect to the maximum net power or minimum levelized cost of electricity (LCOE). Cycle optimization is also included, using a minimum superheat temperature to avoid turbine erosion, which is usually neglected in the literature. The results show that economic optimums are found in the far superheated region, while thermal optimums are obtained with dry saturated or with slightly superheated vapor at the turbine inlet ($\Delta T_{sup}$ < 5 °C). Supercritical cycles achieve better thermal performance than subcritical cycles for $T_{geo}$ = 179–200 °C. Internal heat recuperation improves the cycle performance: the net power output increases and the LCOE decreases, but specific installation costs (SICs) increase due to the additional heat exchanger. For geothermal temperatures of $T_{geo}$ = 120 → 150 °C, the costs are LCOE = 100 → 80 $USD_{2022}$/MWh and SIC = 7000 → 5250 $USD_{2022}$/kW, while for geothermal temperatures of $T_{geo}$ = 150 → 200 °C, the estimated costs are LCOE = 80 → 70 $USD_{2022}$/MWh and SIC = 5250 → 4600 $USD_{2022}$/kW.

**Keywords:** geothermal power plant; thermoeconomic analysis; levelized cost of electricity; specific installation costs; binary cycle optimization

## 1. Introduction

The global installed capacity of geothermal energy for electricity generation has grown steadily over the past decade, from 10.9 GW in 2010 to 14.4 GW in 2021 [1,2]. The annual electricity generation from geothermal plants was 97.2 TWh in 2021 [3], corresponding to 6750 full load hours (with an average capacity factor of 77%). The global installed capacity and annual electricity generation from geothermal sources are expected to exceed 19.0 GW and 130 TWh, respectively, by the end of 2025 [1].

Geothermal power makes a valuable contribution to the energy mix, even though its share is small compared to that of wind or solar power. For comparison, the total global installed capacities of wind and solar power in 2021 were 622.7 GW and 586.4 GW, respectively, with an annual electricity generation of 1429.6 TWh and 724.1 TWh, respectively [4]. Nevertheless, geothermal energy offers sustainable electricity generation without greenhouse gas emissions and at an acceptable cost [5].

The installation cost of geothermal power plants is highly dependent on the depth and properties of the geothermal reservoir, such as the temperature, pressure, mass flow rate, dissolved gases, and solids in the geothermal fluid. The reservoir properties determine the most appropriate technology for a geothermal plant, resulting in a good compromise between the cost of installation and maintenance and the amount of generated electricity. Geothermal power plants come in three main types: (1) flash steam, (2) dry steam, and (3) binary cycle [6]. Compared to flash steam and dry steam systems, binary cycles exploit lower geothermal temperatures by using a working fluid with a lower boiling point in the secondary loop [7]. In general, the specific installation cost of binary cycles is higher than

those of dry steam and flash steam power plants because binary cycles operate at lower geothermal temperatures and achieve lower efficiencies.

Today, engineers and researchers are focused on developing improved components and systems and finding new working fluids for binary cycle configurations capable of utilizing low-temperature heat sources in geothermal and waste heat applications. The performance of binary systems can be evaluated using a variety of approaches: energy and exergy analyses, economic performance, and environmental impact, or a combination of these [8].

Song et al. [9] studied the thermoeconomic performance of subcritical and transcritical geothermal power plants with Organic Rankine Cycle (ORC) technology, considering different working fluids. Of the working fluids studied, R245fa is best suited for dry saturated subcritical and transcritical cycles, while isobutane is the best choice for subcritical superheated cycles. Putera et al. [10] performed a thermoeconomic analysis of a geothermal binary power plant in Indonesia. For a geothermal source temperature of 180 °C and a mass flow rate of 48 kg/s, they estimated the specific installation costs to be between 3000 and 3700 USD/kW with payback periods between 13 and 17 years. Javanshir et al. [11] analyzed the thermoeconomic performance of a combined ORC system for power generation and cooling, using R134a, R22, and R143a as the working fluids. They found that R22 achieved the highest energy efficiency but also the highest unit product cost, while R134a had the lowest energy and exergy efficiency. Overall, the best thermoeconomic performance was obtained with R143a at a unit product cost of 60.7 USD/GJ and with energy and exergy efficiencies of 27.2% and 57.9%, respectively. The thermoeconomic performance of a combined Kalina cycle for power generation and an absorption refrigeration cycle for cooling energy with ammonia–water as the working fluid was studied by Javanshir et al. [12]. The combined system achieved an exergy efficiency of 29.8–34.7% and a product unit cost of 15.0–15.8 USD/GJ, depending on the optimization objective. Güler et al. [13] studied the exergoeconomic performance of a binary dual-pressure system with geothermal brine at 165 °C and 450 kg/s, with n-pentane as the working fluid. They estimated the power generation cost to be between 49 and 58 USD/MWh and the exergy efficiency to be between 39% and 50%. Tagliaferri et al. [14] analyzed the technoeconomic performance of supercritical $CO_2$ cycles for heat and power generation. They estimated the levelized cost of electricity (LCOE) to be between 118 and 169 EUR/MWh, for direct supercritical $CO_2$ cycles combined with isobutane in ORC cycles for electricity generation or with cogeneration for district heating. Toffolo et al. [15] compared the thermoeconomic performance of binary cycles with isobutane and R134a for geothermal source temperatures in the range between 130 and 180 °C. They found that optimum cycles are subcritical for isobutane and supercritical recuperated for 134a.

In general, the literature agrees that geothermal binary cycles perform better when the temperature profiles of the geothermal fluid and the working fluid are closely matched. As a rule of thumb, good thermal matching is achieved when the critical temperature of the working fluid is 20–40 °C lower than the inlet temperature of the geothermal source [16]. For example, R134a, R227ea, and R1234yf are suitable for geothermal temperatures of 90–130 °C, isobutane and n-butane for the range of 140–180 °C, R245fa for temperatures of 180–200 °C, and isopentane for above 200 °C.

Mustapić et al. [17] compared the performance of simple and advanced binary cycles (with two stages and two pressures) for geothermal temperatures in the range of 120–180 °C. At lower source temperatures (120 °C), the highest specific net power was obtained with a two-pressure cycle using R1234yf. At geothermal temperatures around 140 °C, the simple and advanced cycles with using R1234yf and R1234ze achieved comparable results. At temperatures around 180 °C, the single-stage configuration with isobutane and the dual-pressure configuration with n-butane achieved the best results. Prasetyo et al. [18] experimentally investigated the performance of an ORC system using R123 for a low-temperature (120 °C) geothermal well. The ORC device used superheated R123 in a scroll expander and achieved thermal efficiencies in the range of 7.2% to 8.6%. Algieri [19] studied

the thermal performance of subcritical and transcritical binary cycles for a high-temperature (230 °C) geothermal well. Transcritical cycle configurations achieved the highest energy efficiency (17.7%), while isopentane performed better than isobutane and R245ca for the high-temperature geothermal reservoir. Alghamdi et al. [20] studied the energy and exergy performance of a double-flash cycle with zeotropic mixtures for power generation from a high-temperature (200 °C) geothermal source. The highest net power and the lowest exergy dissipation were obtained with a mixture of cyclohexane and R236ea, which provided the best thermal match with the geothermal fluid.

Our literature review shows that the thermoeconomic approach is becoming increasingly important, as it allows us to simultaneously determine the power generation cost and the quality of geothermal energy conversion. The cited authors often report different and contradictory results, even for the same cycle configurations and working fluids. This is due to different assumptions regarding the characteristics of the geothermal reservoir and the operating conditions of the power plant, or different choices of objective functions for cycle optimization. For example, cycle configurations with internal heat recuperation have been found to provide little to no thermal improvement and increase installation costs [9], while other studies have concluded that internal heat recuperation is particularly beneficial for subcritical superheated and supercritical cycles [15]. Most of the cited studies address the thermal performance of different cycle configurations and working fluids in an attempt to improve the thermal match with the heat source. In general, these studies focus on cycle optimization, either from a thermal or economic standpoint, while neglecting the subtle changes around the optimum. In general, supercritical cycles are found to give better thermal performances than subcritical cycles, a conclusion that could shift in favor of subcritical cycles if turbine expansion through the wet steam region and blade erosion are to be avoided. The economic performance of geothermal power plants is evaluated using a variety of cost metrics, with simplified methods for estimating specific installation costs and payback periods more commonly used.

In this study, a comprehensive thermoeconomic approach is developed to evaluate the performance of subcritical and supercritical cycles in the geothermal temperature range between 100 and 200 °C. The working fluid in the Rankine cycle is isobutane with dry saturated or superheated conditions at the turbine inlet. The analysis is extended to cycle configurations with and without internal heat recovery and to isobutane conditions with the minimum superheat required to avoid droplet formation and turbine blade erosion. The objective functions for cycle optimization are maximum net power output and minimum levelized cost of electricity (LCOE). The LCOE and specific installation cost (SIC) are the preferred cost metrics in this study because they provide clear information about the cost-effectiveness of geothermal electricity and allow us to directly compare our results with those of other renewable energy projects.

## 2. Materials and Methods

### 2.1. The Rankine Cycle Configuration

The geothermal power plant consists of a Rankine cycle in a single-pressure configuration with isobutane as the working fluid, as shown in Figure 1. A thermoeconomic analysis is performed for a geothermal heat source with a mass flow rate of 225 kg/s (810 t/h) and inlet temperatures between 100 °C and 200 °C. The geothermal fluid is modeled as water using the IAPWS-IF97 formulation for the thermodynamic and transport properties of fluids contained in the CoolProp database. After the production well, the geothermal fluid, state ⑨, transfers thermal energy to the ORC fluid (isobutane) in the evaporator, state change ⑨ → ⑩, and in the preheater, ⑩ → ⑪. Geothermal brine is reinjected into the geothermal reservoir with a minimum temperature of 70 °C, state ⑪, to prevent scaling on heat transfer surfaces and silica precipitation. In the ORC loop, dry saturated or superheated isobutane expands in the turbine from ① to ②. Isobutane is a dry working fluid, and its expansion ends in the superheated region. The sensible heat content from the turbine exhaust vapors, which would otherwise be discharged by the condenser, is

recovered in the heat recuperator (desuperheater), state change ② → ③. Isobutane vapors are cooled and liquefied in the condenser, ③ → ④ → ⑤. Waste heat is discharged into the atmosphere by forced convection cooling towers. The cooling water temperature increases from ⑫ to ⑬ across the condenser. The liquid condensate is pumped by feed pumps, ⑤ → ⑥, and it is preheated in the heat recuperator, ⑥ → ⑦. Further heating of the liquid phase takes place in the preheater up to the boiling point, ⑦ → ⑧, while evaporation and superheating takes place in the evaporator ⑧ → ①. The thermodynamic and transport properties of isobutane are calculated using CoolProp functions [21].

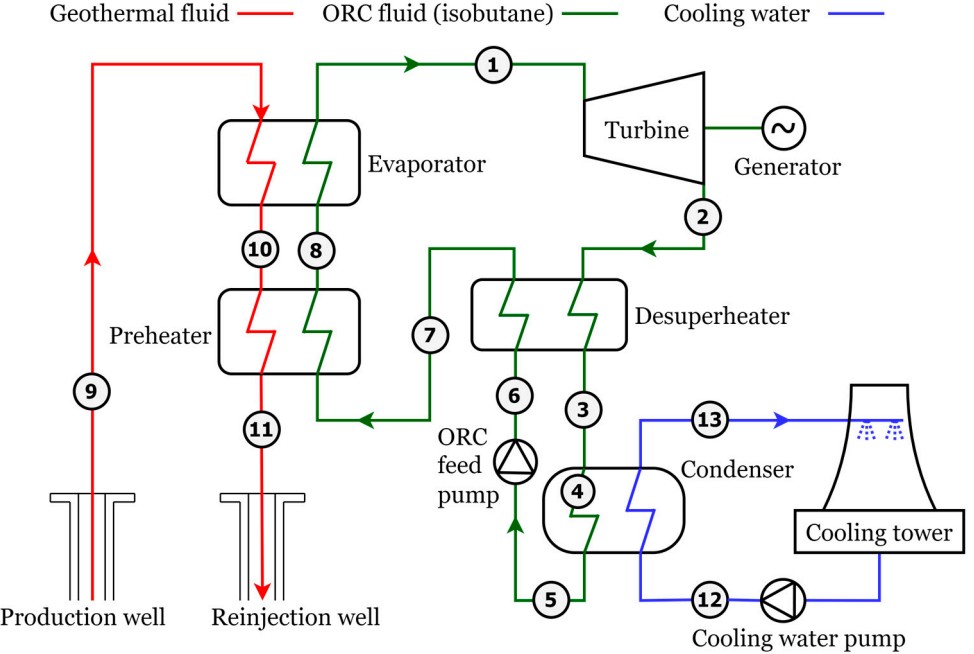

**Figure 1.** The Rankine cycle configuration.

### 2.2. The Thermodynamic Model

The thermodynamic model applies mass and energy conservation equations to the particular components of the binary geothermal power plant. The heat balance equations for the evaporator and the preheater consider the heat flow rates from the geothermal fluid to the working fluid, that is:

$$\dot{m}_{ORC}(h_1 - h_8) = \dot{m}_{GTF}(h_9 - h_{10})\eta_{EV} \tag{1}$$

$$\dot{m}_{ORC}(h_8 - h_7) = \dot{m}_{GTF}(h_{10} - h_{11})\eta_{PH} \tag{2}$$

The mass flow rates of the working fluid and geothermal brine are denoted with $\dot{m}_{ORC}$ and $\dot{m}_{GTF}$, while the specific enthalpies are $h_i$. Heat losses from the preheater and evaporator are assumed as 10% [19] ($\eta_{PH} = 0.90$, $\eta_{EV} = 0.90$). The temperature profiles obtained by solving Equations (1) and (2) must respect two minimum temperature conditions, which are considered decision variables in the present work. The minimum reinjection temperature of the geothermal fluid is $T_{11min} = 70\,°C$ and the minimum pinch point temperature difference (PPTD) between the heat source and the working fluid is $\Delta T_{pp,min} = 10\,°C$. In case the calculated reinjection temperature is lower than the minimum of 70 °C, the PPTD must be increased. Heat losses from the heat recuperator are assumed as 5% ($\eta_{RE} = 0.95$). The heat balance equations for the heat recuperator and the condenser are as follows:

$$\dot{m}_{ORC}(h_2 - h_3)\eta_{RE} = \dot{m}_{ORC}(h_7 - h_6) \tag{3}$$

$$\dot{m}_{\text{ORC}}(h_3 - h_5) = \dot{m}_{\text{CW}}(h_{13} - h_{12}) \tag{4}$$

Equations (3) and (4) are solved assuming a minimum PPTD of 10 °C between the vapor phase and the liquid phase in the heat recuperator and 10 °C between the saturation temperature of the condenser and the cooling water. The condenser saturation temperature (and pressure) is determined assuming a cooling water temperature increase of $T_{13}$–$T_{12}$ = 7 °C and a cooling tower efficiency of $\eta_{\text{CT}}$ = 0.75. The cooling tower efficiency is calculated as the ratio between the cooling water temperature range and the maximum cooling water temperature approach as follows:

$$\eta_{\text{CT}} = \frac{T_{13} - T_{12}}{T_{13} - T_{\text{wb}}} \tag{5}$$

The air wet bulb temperature ($T_{\text{wb}}$) is calculated for an air temperature of 20 °C and a relative humidity of 50%. The turbine power output and the gross power output of the geothermal power plant are calculated using the following expressions:

$$\dot{W}_{\text{T}} = \dot{m}_{\text{ORC}}(h_1 - h_2) = \dot{m}_{\text{ORC}}(h_1 - h_{2s})\eta_{\text{T}} \tag{6}$$

$$\dot{W}_{\text{gross}} = \left(\dot{W}_{\text{T,ORC}} + \dot{W}_{\text{T,NCG}}\right)\eta_m \eta_{\text{el}} \tag{7}$$

The isentropic efficiency of the turbine is $\eta_{\text{T}}$ = 0.88. The power plant gross output is calculated from the turbine power output, assuming mechanical and electrical losses of $\eta_m$ = 0.98 and $\eta_{\text{el}}$ = 0.98, respectively. The power consumption of the feed pumps and of the cooling water pumps are calculated as follows:

$$\dot{W}_{\text{FP}} = \frac{\dot{m}_{\text{ORC}}(h_{6s} - h_5)}{\eta_{\text{FP}}} \tag{8}$$

$$\dot{W}_{\text{CWP}} = \frac{\dot{m}_{\text{CW}}\, g\, H}{\eta_{\text{CWP}}} \tag{9}$$

The isentropic efficiency of the pumps is assumed as $\eta_{\text{FP}} = \eta_{\text{CWP}}$ = 0.75 and the water head in the cooling system is $H$ = 30 m. Auxiliary power consumption (cooling tower fans, control and regulation, downhole pump) is assumed to be 5% ($\dot{W}_{\text{AUX}} = 0.10 \times \dot{W}_{\text{gross}}$) of the gross power output [22]. The net power output of the geothermal power plant is then as follows:

$$\dot{W}_{\text{net}} = \dot{W}_{\text{gross}} - \left(\dot{W}_{\text{FP}} + \dot{W}_{\text{CWP}} + \dot{W}_{\text{AUX}}\right) \tag{10}$$

The net thermal efficiency of the binary cycle is calculated as the ratio between the net power output and the total heat flow rate in the preheater and evaporator:

$$\eta_{\text{th,net}} = \frac{\dot{W}_{\text{net}}}{\dot{Q}_{\text{PR}} + \dot{Q}_{\text{EV}}} = \frac{\dot{W}_{\text{net}}}{\dot{m}_{\text{GTF}} \cdot (h_9 - h_{11})} \tag{11}$$

In the above equation, the reinjection enthalpy depends on the quality of thermal matching between the heat source and the working fluid, as well as on the general operating conditions of the cycle. Regardless of the reinjection enthalpy, the absolute net efficiency compares the net power output to the maximum available geothermal heat flow rate that is exchanged when the minimum reinjection temperature is achieved ($T_{11\text{min}}$ = 70 °C)

$$\eta_{\text{net,abs}} = \frac{\dot{W}_{\text{net}}}{\dot{m}_{\text{GTF}} \cdot (h_9 - h_{11\text{min}})} \tag{12}$$

### 2.3. The Heat Transfer Model

The heat exchangers (preheater, evaporator, condenser, and recuperator) are modeled as shell-and-tube heat exchangers. The heat transfer area must be estimated correctly because it has a large impact on the total installation cost and the cost of electricity generation [6]. The heat transfer area in each of the heat exchangers is determined as follows:

$$A_s = \frac{\dot{Q}}{U \cdot \Delta T_{lm}} \tag{13}$$

where $\dot{Q}$ is the heat flow rate in the heat exchanger determined from the heat balance Equations (1)–(4), while $U$ and $\Delta T_{lm}$ are the overall heat transfer coefficient and the log mean temperature difference calculated using the following expressions:

$$U = d^{-1} \left( \frac{1}{h_i \cdot d_i} + \frac{1}{2\lambda_s} \ln \frac{d_o}{d_i} + \frac{1}{h_o \cdot d_o} \right)^{-1} \tag{14}$$

$$\Delta T_{lm} = \frac{\Delta T_i - \Delta T_o}{\ln \frac{\Delta T_i}{\Delta T_o}} \tag{15}$$

The temperature difference between the two fluids in heat exchange is denoted by $T_i$ and $T_o$ for the inlet and outlet sides of the heat exchanger. In the preheater and evaporator, the geothermal fluid flows inside the tubes, while isobutane flows on the shell side. In the condenser, the cooling water flows inside the tubes and the isobutane condenses on the shell side. In the recuperator, the liquid flows inside the tubes while the vapor flows on the shell side. The heat transfer coefficient for single-phase turbulent flow on the tube side is determined using the Gnielinski correlation [23] as follows:

$$Nu = \frac{h_i \cdot d_i}{k} = \frac{(f/8)(Re - 1000)Pr}{1 + 12.7(f/8)^{1/2}(Pr^{2/3} - 1000)} \tag{16}$$

$$f = [0.79 \ln(Re) - 1.64]^{-2} \tag{17}$$

The heat transfer coefficient for shell-side single-phase flow is evaluated using the Zukauskas correlation [23] for crossflow over staggered tube banks:

$$Nu = \frac{h_o \cdot d_o}{k} = C\, Re^x Pr^{0.36} (Pr/Pr_w)^{0.25} \tag{18}$$

Table 1 lists the values for the constant $C$ and the Reynolds number exponent $x$, which depend on the Reynolds number range, as well as on the transverse ($X_T$) and longitudinal ($X_L$) tube pitches in the tube bank [23,24]. The fluid properties are evaluated using the arithmetic mean between the inlet and the outlet temperature, except for the Prandtl number $Pr_w$, which is evaluated at the tube wall temperature.

**Table 1.** Parameters for the Zukauskas correlation (18).

| Reynolds Number Range | Constant $C$ | Exponent $x$ |
|---|---|---|
| 0–500 | 1.04 | 0.4 |
| 500–1000 | 0.71 | 0.5 |
| $1000$–$2 \times 10^5$ | $0.35\,(X_T/X_L)^{0.2}$ | 0.6 |
| $2 \times 10^5$–$2 \times 10^6$ | $0.031\,(X_T/X_L)^{0.2}$ | 0.8 |

Two-phase heat transfer occurs on the shell-side of the evaporator and condenser. The heat transfer coefficient in the evaporator is evaluated using the Cooper correlation [24] for nucleate boiling over horizontal tube banks:

$$h_{\mathrm{nb}} = \left[ 55(T_{\mathrm{w}} - T_{\mathrm{sat}})^{0.67} P_r^{0.12} \log\left(1/P_r\right)^{-0.55} M^{-0.5} \right]^{1/0.33} \tag{19}$$

In the above equation, $P_r$ is the reduced pressure calculated as the ratio between the saturation pressure and the critical pressure ($P_r = P_{\mathrm{sat}}/P_{\mathrm{cr}}$), $M$ is the molecular weight, and ($T_{\mathrm{w}} - T_{\mathrm{sat}}$) is the difference between the tube wall temperature and the saturation temperature. For film condensation over horizontal tube bundles, the Rose correlation [23,24] is used to calculate the heat transfer coefficient of the condenser.

$$h_{\mathrm{fc}} = 0.728 \left[ \frac{g \cdot \rho_{\mathrm{L}}(\rho_{\mathrm{L}} - \rho_{\mathrm{V}})(h_{\mathrm{V}} - h_{\mathrm{L}})\lambda_{\mathrm{L}}^3}{\mu_{\mathrm{L}} \cdot d_{\mathrm{o}} \cdot N^{2/3} \cdot (T_{\mathrm{sat}} - T_{\mathrm{w}})} \right]^{1/4} \tag{20}$$

In the above equation, $g = 9.81$ m/s$^2$ is the standard acceleration due to gravity and $d_{\mathrm{o}}$ is the tube outside diameter. The physical properties of the liquid phase are density $\rho_{\mathrm{L}}$, enthalpy $h_{\mathrm{L}}$, thermal conductivity $\lambda_{\mathrm{L}}$, and dynamic viscosity $\mu_{\mathrm{L}}$, while the the properties of the vapor phase are density $\rho_{\mathrm{V}}$ and enthalpy $h_{\mathrm{V}}$. In horizontal tube bank designs, the lower tubes have an increased layer thickness due to condensate drainage from the upper tubes. The correction for the reduced heat transfer coefficient at the lower tubes is accounted for by the number of tubes stacked on top of each other ($N$).

*2.4. The Economic Model*

The present analysis uses the discounted cash flow (DCF) method for calculating the levelized cost of electricity (LCOE) of the binary geothermal power plant [3]. This method estimates the net present value of the investment by calculating the future value of the revenues. The LCOE formula is as follows:

$$LCOE = \frac{C_{\mathrm{CAP}} + \sum_{t=1}^{t=N} (C_{\mathrm{O\&M}} + C_{\mathrm{M}})_t \frac{(1+e)^t}{(1+r)^t}}{\sum_{t=1}^{t=N} E_{\mathrm{net}}(1-d)^t} \tag{21}$$

The above equation calculates the levelized cost of electricity generation in present-day real USD, including inflation. Total costs, including capital costs ($C_{\mathrm{CAP}}$), operation and maintenance costs ($C_{\mathrm{O\&M}}$), and material costs ($C_{\mathrm{M}}$), are summed accounting for their net present value. The present value of capital costs represents the sum of annuities for a bank loan necessary to finance the initial capital expenditure of the geothermal project. The real discount rate is $r = 5\%$ (weighted average cost of capital: WACC) and the lifetime of the geothermal power plant is $T = 25$ years. It should be noted that the discount rate of 5% (real WACC value) is valid mostly in OECD countries and China, while in the rest of the world, a discount rate of 7.5% would be a better representation [3]. The annual electricity generation is calculated assuming $N_{\mathrm{h}} = 7000$ h of full load operating hours. This value reflects the global weighted capacity factor in the range between 75% and 91% for geothermal power plants commissioned in 2021, with an average capacity factor of 80% [3]. The annual capacity degradation rate is assumed as $d = 1\%$. The capacity degradation rate accounts for the reduction in full load hours arising from the thermal depletion of the geothermal reservoir [25] along with more frequent maintenance, inspections, and repair works over the power plant's lifetime. Operation and maintenance (O&M) costs are assumed at an initial value of 135 USD/kW/year [3]. Material costs comprise isobutane and cooling water losses due to leakages and evaporation in the cooling towers. The price escalation rate for future material and maintenance costs is assumed as $e = 2\%$, which arises from a more frequent fouling, corrosion, and wearing over the power plant's lifetime [26,27]. All cost metrics in the present study are reported in real USD as of 2022.

Table 2 summarizes the assumptions for the LCOE calculations. In general, geothermal power is capital intensive, and installation costs are highly site specific. The characteristics of the geothermal reservoir, the cost of site exploration and drilling, the number and

depth of wells, and the type of power plant technology all affect the cost of geothermal electricity [28]. The macroeconomic environment and the economy of scale also play a role in the final cost of geothermal electricity [27]. While all of these factors can hardly be captured by a single cost model, the objective of this study is rather to quantify the LCOE of the average binary geothermal power plant and to understand how operating conditions can be optimized to achieve minimum LCOE.

**Table 2.** Assumptions for the LCOE.

| Cost Parameter | Value | Ref. |
|---|---|---|
| Discount rate (real value), $r$ | 5.0% | [3] |
| Price escalation rate, $e$ | 2% | [26,27] |
| Capacity degradation rate, $d$ | 1% | [25] |
| Project lifetime, $T$ | 25 years | [3] |
| Contingencies and fees, $f_{TM}$ | 18% | [29] |
| Auxiliary costs, $f_{AUX}$ | 50% | [29] |
| Annual O&M costs | 135 USD/kW | [3] |

Total capital costs are estimated from the cost of major equipment using multiplication factors. This method was introduced by Lang [30,31] and uses multiplication factors to account for direct costs (equipment purchase and installation, piping, insulation and fire protection, electrical work, instrumentation and controls, site preparation) and indirect costs (transportation, contingencies, fees). This method has since been revised with the goal of more accurate cost estimates [32,33]. The approach developed by Turton et al. [29] is used in this study because it provides a good estimate of the capital cost in greenfield renewable energy projects:

$$C_{GRP} = \sum_{i}^{N} C_{EQ,i} [f_{TM} f_M f_P + f_{aux}] \left( \frac{CI_{2022,i}}{CI_{ref,i}} \right) \qquad (22)$$

In Equation (22), the purchased equipment costs are denoted by $C_{EQ,i}$ and depend on project-specific variables and operating conditions. These costs are extrapolated from cost correlations found in the literature developed for reference power plant size and baseline conditions. The equipment costs are updated with the corresponding cost index ratios ($CI_{2022}/CI_{ref}$), which compare the average cost indices in 2022 with the cost indices in the years in which the correlations were published [34]. Table 3 shows the cost correlations for the main equipment of the binary geothermal power plant.

**Table 3.** Cost correlations for the purchased equipment.

| Equipment Type | Size Unit | Cost Correlation ($C_{EQ,i}$ in USD) and Literature Source | Reference Cost Index ($CI_{ref}$) |
|---|---|---|---|
| Turbine | Power (kW) | $C_{EQ} = 1900 \dot{W}_T^{0.75} - 14000$; [35] | 631.8 (CE: equipment, 2010) |
| Shell-and-tube heat exchanger | Heat transfer area, $A$ (m$^2$) | $C_{EQ} = -0.06395 A^2 + 947.2 A + \log A + 227.9$; [36] | 614.5 (CE: heat exchangers, 2020) |
| Centrifugal pump | Power (kW) | $C_{EQ} = -0.03195 \dot{W}_P^2 + 467.2 \dot{W}_P + \log \dot{W}_P + 20480$; [36] | 1084.3 (CE: pumps, 2020) |
| Generator | Power (kW) | $C_{EQ} = 2775000 \left( \dot{W}_G / 11800 \right)^{0.94}$; [15] | 511.3 (CE: electrical equipment, 2012) |
| Cooling tower | Water flow rate (l/s) | $C_{EQ} = 1500 \dot{V}_{CW}^{0.9} + 170000$; [35] | 631.8 (CE: equipment, 2010) |
| Isobutane | Mass (kg): 7 kg/kW$_G$ | $C_{EQ} = 1.43 \left( 7 \cdot \dot{W}_G \right)$; [37] | 1036.9: (CE: equipment, 2022) |

The baseline cost correlations are corrected with material ($f_M$) and pressure ($f_P$) factors if the equipment must withstand adverse operating conditions and corrosive fluids. In this study, a material factor for stainless steel ($f_M = 1.5$) is assumed for the pumps and the turbine, while the cost correlation for heat exchangers has already been developed for carbon steel shell and stainless-steel tubes, and a material factor is not required ($f_M = 1.0$) [36]. Table 4 summarizes the material factors ($f_M$) and the pressure factor ($f_P$), which is calculated using the following correlation [29]:

$$\log f_P = C_1 + C_2 \log(p) + C_3 \log(p)^2 \tag{23}$$

**Table 4.** Correction factors for materials and operating pressures.

| Correction Factor | Range/Type | Value |
|---|---|---|
| Material type, $f_M$ | Stainless steel (SS) | 1.5 |
| Pressure factor constants for heat exchangers, $f_P$ (20) | $p < 5$ bar | $C_1 = C_2 = C_3 = 0$ |
| | $5 < p < 140$ bar | $C_1 = 0.03881$ $C_2 = -0.11272$ $C_3 = 0.08183$ |
| Pressure factor constants for centrifugal pumps, $f_P$ (20) | $p < 10$ bar | $C_1 = C_2 = C_3 = 0$ |
| | $10 < p < 100$ bar | $C_1 = -0.3935$ $C_2 = 0.3957$ $C_3 = -0.00226$ |
| Contingencies and fees, $f_{TM}$ Auxiliary costs, $f_{AUX}$ | Greenfield projects | 1.18 0.5 |

The equipment costs multiplied by correction factors for pressure and material are called bare module costs ($C_{BM} = C_{EQ} \times f_M \times f_P$), which include all direct and indirect costs. The total module cost is obtained by adding the contingencies and fees to the bare module cost ($C_{TM} = C_{BM} \times f_{TM}$). Contingency costs and fees are assumed as 15% and 3% [29], which means that a total module factor of $f_{TM} = 1.18$ is used in (22). In the case of greenfield projects, the costs of the purchased equipment are increased by 50% to account for auxiliary costs. These auxiliary costs include the costs of site development and the construction of auxiliary buildings and utilities. The multiplication factor for auxiliary costs is assumed as $f_{aux} = 0.50$.

The present value of total capital costs is the product of the project lifetime and the annuity ($C_{CAP} = A_{CAP} \times T$), which is constant over the project lifetime and calculated as follows:

$$A_{CAP} = \frac{C_{GRP} \cdot r}{1 - (1+r)^{-T}} \tag{24}$$

The purchase cost of the working fluid is included in the equipment costs (22), assuming a specific quantity of isobutane per unit of net power of 7.0 kg/kW [37]. Fugitive emissions of isobutane are assumed as 0.5% at the annual basis [37]. The average price of isobutane in 2022 was 1.43 USD/kg, as reported by the Nasdaq database [38]. Evaporation losses from cooling towers are estimated assuming a specific loss of 0.2% of the water mass flow rate for each 1 °C decrease in temperature in the cooling towers [39]. The price of makeup water is 20 USD/1000 m$^3$ [33]. The costs of makeup water and isobutane are included in the material costs of Equation (21). Specific installation costs (SIC) are determined as the ratio between the total investment costs (Equation (22)) and the generated net power output (Equation (10)) in the geothermal project:

$$SIC = \frac{C_{GRP}}{\dot{W}_{net}} \tag{25}$$

## 3. Results and Discussion

### 3.1. Validation of the Thermodynamic Model

The results obtained in the present study are compared with results from the literature. To make a fair comparison, the referenced studies were thoroughly reviewed, and all decision variables were included in the present model. The decision variables generally include geothermal mass flow rate and temperature, minimum reinjection temperature, minimum PPTD, and the working fluid pressure and temperature at the turbine inlet. The outputs include the working fluid mass flow, gross and net power output, net thermal efficiency, and power consumption in feed pumps and cooling tower fans. The present thermodynamic model accurately predicts the results reported in [15], as can be seen in Table 5. Minor discrepancies can be attributed to the use of different databases for the physical properties of isobutane and different approaches for evaluating pressure drops and heat losses in the piping and heat exchangers. In Nevada (USA), the Tungsten Mountain geothermal power plant [40] uses binary cycle technology with a single Ormat Energy Converter for a nameplate capacity of 37 MW. The geothermal fluid flows from four production wells at a mass flow rate of $4 \times 250$ kg/s and with a temperature of 142 °C. The average net power output is 24–27 MW, and the parasitic load is 10–15% [40]. The present model puts the maximum net power output at 25.82 MW, obtained with subcritical non-recuperated isobutane and a dry saturated state at the turbine inlet ($p_1 = 16.5$ bar, $T_1 = 90.3$ °C). The estimated total parasitic load is 4.37 MW (2.15 MW in the feed pumps and 2.22 MW in the cooling water pumps) or 14.47% of the gross power output. These results are obtained for average ambient air conditions: a temperature of 20 °C and a relative humidity of 50%.

**Table 5.** Validation of the present thermodynamic model.

| Decision Variable | Value | | | | | |
|---|---|---|---|---|---|---|
| Geothermal fluid inlet temperature, °C | 130 | | | 170 | | |
| Geothermal fluid return temperature, °C | >70 | | | >70 | | |
| Geothermal fluid mass flow rate, kg/s | 100 | | | 100 | | |
| Turbine inlet temperature, °C | 84.4 | | | 135.1 | | |
| Maximum ORC cycle pressure, bar | 14.3 | | | 35.2 | | |
| Condenser saturation temperature, °C | 32.8 °C | | | 33 °C | | |
| Pinch point temperature difference, °C | 10 | | | 10 | | |
| Efficiency: turbine/pumps/generator, - | 0.85/0.70/0.96 | | | 0.85/0.70/0.96 | | |
| Results comparison | Present model | Reference data [15] | % diff. | Present model | Reference data [15] | % diff. |
| ORC fluid mass flow rate | 63.3 kg/s | 62.4 kg/s | 1.44 | 104.17 kg/s | 105.6 kg/s | −1.35 |
| Gross power output | 2.36 MW | 2.39 MW | −1.26 | 6.53 MW | 6.61 MW | −1.21 |
| Net power output | 1.77 MW | 1.81 MW | −2.21 | 5.02 MW | 5.08 MW | −1.18 |
| Auxiliary power (feed pump, fans) | 0.59 MW | 0.58 MW | 1.72 | 1.51 MW | 1.53 MW | −1.31 |
| Net thermal efficiency | 7.24% | 7.48% | −3.21 | 11.80% | 11.94% | −1.17 |

### 3.2. Validation of the Economic Model

The results of the present study are compared with the cost breakdown for the Stillwater geothermal power plant in Nevada, USA [15]. This power plant uses isobutane at a maximum pressure of 30 bar for a total net power output of 33.6 MW. The power plant was completed in 2009 for a total cost of 132.9 million USD$_{2009}$ and was later upgraded to a hybrid power plant using geothermal, solar thermal, and PV energy [41,42]. The costs of the Stillwater power plant are updated using a cost index ratio of 1.69, which arises from the chemical engineering indices between 2022 and 2009 ($CI_{2022}/CI_{2009} = 1036.9/613.2 = 1.69$). Table 6 gives a comparison between the costs predicted by the present model and those published in [15]. The present model reproduces the equipment and total module costs fairly well, with differences of 10.38% and 1.11%, respectively. The total greenfield costs, on the other hand, differ by −26.11%. Possible reasons for this discrepancy include the

long period between the published data costs and present-day costs and the properties of the Stillwater geothermal site. Generally, cost updating becomes less accurate after 5 years [32], while the site properties can be accounted for by adjusting the auxiliary costs. The present economic model assumes auxiliary costs to be 50% of the equipment costs ($f_{aux}$ = 0.5 in Equation (22)), but values in the range between 20% and 100% were reported in [29], depending on project circumstances.

**Table 6.** Validation of the present economic model.

| Cost Type | Present Model USD$_{2022}$ | Reference Data [15] USD$_{2009}$ | USD$_{2022}$ | Difference |
|---|---|---|---|---|
| Equipment costs, $\times 10^6$ | 83.48 | 44.75 | 75.63 | 10.38% |
| Total module costs, $\times 10^6$ | 124.19 | 72.67 | 122.82 | 1.11% |
| Greenfield project costs, $\times 10^6$ | 165.93 | 132.88 | 224.57 | −26.11% |
| Specific installation costs, USD/kW | 4938 | 3955 | 6684 | −26.11% |

The present economic model estimates specific installation costs (SICs) in the range between 4100 and 5200 USD$_{2022}$/kW for geothermal power plants with subcritical isobutane and heat recuperation. The lower end of the cost range refers to power plants with a design net output of 50 MW and a geothermal source temperature of 180 °C. The higher end of the cost range is for geothermal power plants with net design output of 5 MW and a geothermal source temperature of 140 °C. These specific costs are comparable with those given in the IRENA-2021 report for renewable power costs [3], which estimates that most binary geothermal power plants are installed for 3300–6000 USD$_{2022}$/kW, with an average SIC of 4700 USD$_{2022}$/kW. Lazard's report [43] puts the SIC between 4700 and 6075 USD/kW. Lemmens [32] surveyed the literature and concluded that the average SICs for geothermal binary units are 3156 EUR$_{2014}$/kW, which is updated to 6000 USD$_{2022}$/kW, using an exchange rate of 1.33 USD$_{2014}$/EUR$_{2014}$ and a cost index ratio of $CI_{2022}/CI_{2014}$ = 1036.9/699.4 = 1.48. Sanyal [27] suggested a simple correlation for the SICs of geothermal power plants: $SIC = 2500 \times \exp[-0.003(\dot{W} - 5)]$, with results in USD$_{2004}$/kW and power plant sizes ($\dot{W}$) in MW. The chemical engineering index nearly doubled from 2004 to 2022 ($CI_{2022}/CI_{2004}$ = 1036.9/536.9 = 1.93), so that the updated Sanyal correlation is $SIC = 4800 \times \exp[-0.003(\dot{W} - 5)]$. This updated correlation obtains SICs between 4200 and 4800 USD$_{2022}$/kW for power plant sizes from 50 to 5 MW. Lawless et al. [44] reported an average capital cost of around 4500 USD/kW for geothermal power plants using medium enthalpy sources in New Zealand.

According to the IRENA-2021 report [3], the global average levelized cost of electricity (LCOE) was 68 USD/MWh for geothermal power plants completed in 2021. The LCOE ranged between 37 USD/MWh for power plant upgrades and extensions and 170 USD/MWh for smaller power plants on new geothermal fields in remote locations. The Lazard's annual report [43] estimates the LCOE of geothermal power in the range between 61 and 102 USD/MWh, while Robins et al. [45] reports values in the range between 67.5 and 74.0 USD/MWh. The cost data from real geothermal power plants have been found in research articles [40], corporate reports [46,47], and energy news websites [48,49]. Figure 2 shows a comparison between model predictions and the real cost data. All costs in Figure 2 are reported in real USD as of 2022.

The present model captures the general trend between the geothermal power costs and the binary plant size. As expected, the real cost data in the graphs are highly scattered because different sources often use different cost methodologies for reporting geothermal power plant data. Some sources report capital expenditures per unit of net output, while others report costs per nameplate capacity, which is then reflected in the LCOE estimates. In addition, cost data are also strongly influenced by geothermal site characteristics and location, the power plant type and technology, and the macroeconomic environment. In

light of this, the results of the present model should be considered as average results for the binary geothermal technology under average site and operating conditions. The present economic model predicts the LCOE in the range between 65 and 100 USD2022/MWh. A geothermal power plant with a design net output of 50 MW and a source temperature of 180 °C achieves an LCOE as low as 67 USD2022/MWh. A smaller power plant (5 MW) using a source temperature of 140 °C achieves an LCOE of 80 USD2022/MWh.

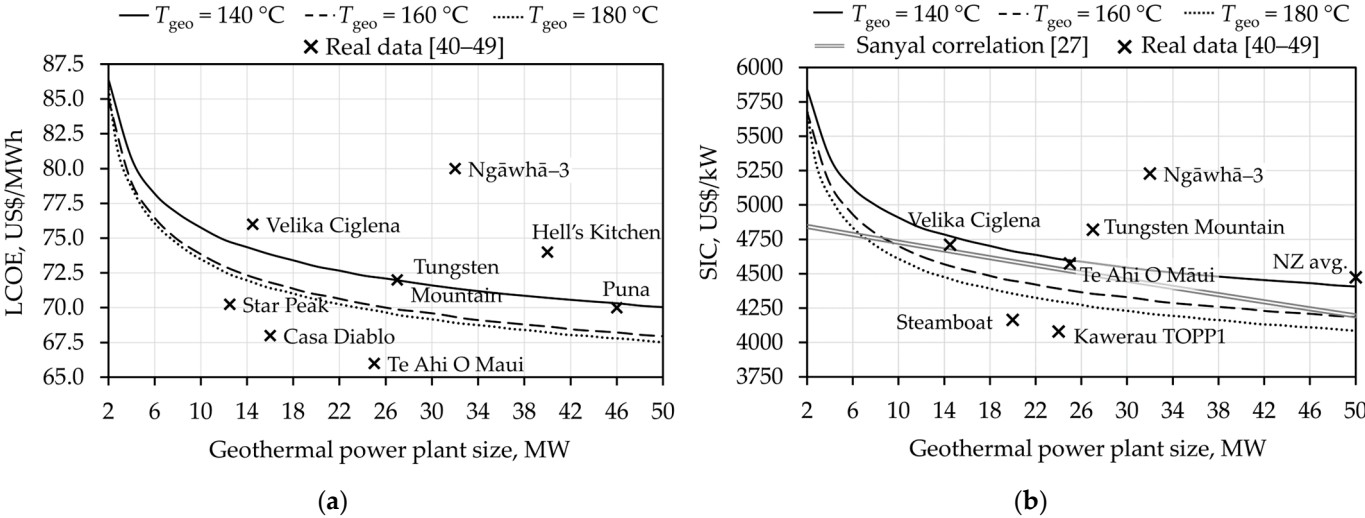

**Figure 2.** Geothermal power costs versus binary plant size: (**a**) levelized cost of electricity (LCOE) and (**b**) specific installation costs (SICs). Comparison between model predictions and real data.

### 3.3. Thermoeconomic Analysis

#### 3.3.1. Subcritical Saturated Isobutane Cycles

Figure 3 shows how the geothermal well temperature and the turbine inlet pressure affect the net power output and the LCOE. For a given turbine inlet pressure, the net power output increases with the geothermal source temperature. This is expected because higher inlet temperatures increase the heat content of the geothermal fluid. The net power curves show a slope change between geothermal temperatures of 150 °C and 170 °C. This is due to temperature constraints on the minimum reinjection temperature and minimum pinch point temperature difference (PPTD). Before the slope change, the net power increases rapidly until when the reinjection temperature reaches the minimum of 70 °C. A further increase in the geothermal well temperature results in a slower increase in the net power, as the PPTD must be increased above 10 °C to maintain the reinjection temperature above 70 °C.

For geothermal temperatures lower than 140 °C, the maximum net power output is obtained with lower pressures (10–15 bar). For geothermal temperatures above 165 °C, the maximum net output is achieved with 30–35 bar, close to the critical pressure of isobutane ($p_{cr}$ = 36.3 bar). For geothermal temperatures above 150 °C, the economic analysis reveals that cycles with turbine inlet pressures under 15 bar result in an LCOE higher than 90 USD/MWh, while turbine inlet pressures above 20 bar achieve an LCOE under 80 USD/MWh.

Figure 3 suggests that for a given geothermal temperature, there is an optimum subcritical isobutane cycle leading to a maximum net power output, and there is another subcritical cycle that ensures a minimum LCOE. Cycle optimization was performed for the maximum net power output or for the minimum levelized cost of electricity. The cycle optimization uses a brute force approach, in which the turbine inlet temperatures and pressures are varied until the thermal optimum points (the maximum net power output) and economic optimum points (the minimum LCOE) are found. The cycle optimization method could be accelerated using artificial intelligence approaches, including fuzzy control systems [50], neural networks, and machine learning [51].

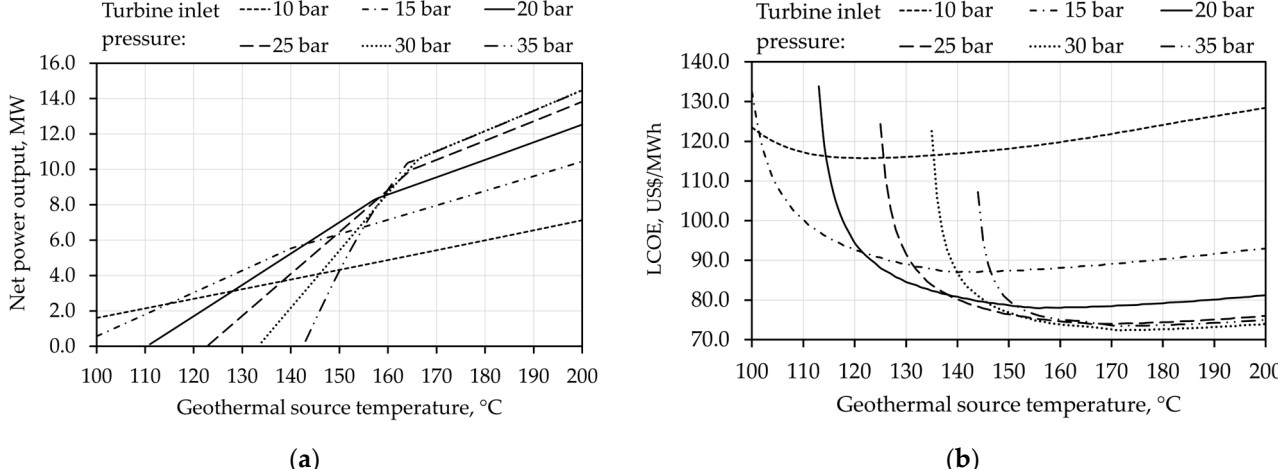

**Figure 3.** Thermoeconomic performance of the subcritical cycle with dry saturated isobutane and for different geothermal source temperatures and turbine inlet pressures: (**a**) net power output and (**b**) levelized cost of electricity (LCOE). Results for a geothermal mass flow rate of 225 kg/s.

The thermal-optimum and cost-optimum turbine inlet pressures are plotted in Figure 4. Three different zones are identified. In the first zone, for geothermal source temperatures under 158 °C, the cost optimum pressure is higher than the thermal-optimum pressure. In the second zone, for geothermal temperatures between 158 °C and 180 °C, the cost-optimum pressure is lower than the thermal-optimum pressure. In the third zone, for geothermal source temperatures above 180 °C, the cost-optimum pressure matches the thermal-optimum pressure of 31.5 bar.

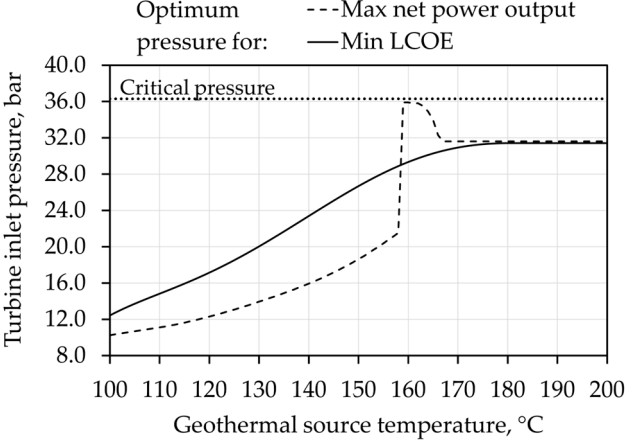

**Figure 4.** Thermal- and cost-optimum turbine inlet pressures for subcritical dry saturated isobutane cycles. Results for a geothermal mass flow rate of 225 kg/s.

The subcritical cycle performance with a geothermal source temperature of 158 °C is further explored for three turbine inlet pressures: 21.6 bar and 36 bar, which compete for the maximum net power output, and 28.9 bar (the minimum LCOE). Figure 5 compares the temperature–entropy charts of these three cycles and the thermoeconomic performance is reported in Table 7. The points in the *T-s* charts are labeled as in the scheme of Figure 1. The cycle with 21.6 bar at the turbine inlet achieves a reinjection temperature of 72.50 °C, which represents the best thermal match among the three compared cycles. An isobutane cycle can be considered an optimum thermal match with the geothermal fluid when the reinjection temperature reaches the minimum value of 70 °C, and the PPTD reaches the minimum value of 10 °C. These two temperature conditions are decision variables in the present study, and the values assumed here are generally recommended in the literature [15,19].

Optimum thermal matching ensures that the total heat flow rate transferred to the working fluid is maximal. In the subcritical cycle with $p_1$ = 21.6 bar, the total transferred heat flow rate (81.53 MW) and the isobutane mass flow rate (198.97 kg/s) are the largest among the three cycles. On the other side, the condenser heat flow rate (67.36 MW) is also the largest, causing an increased heat transfer surface area and higher equipment costs.

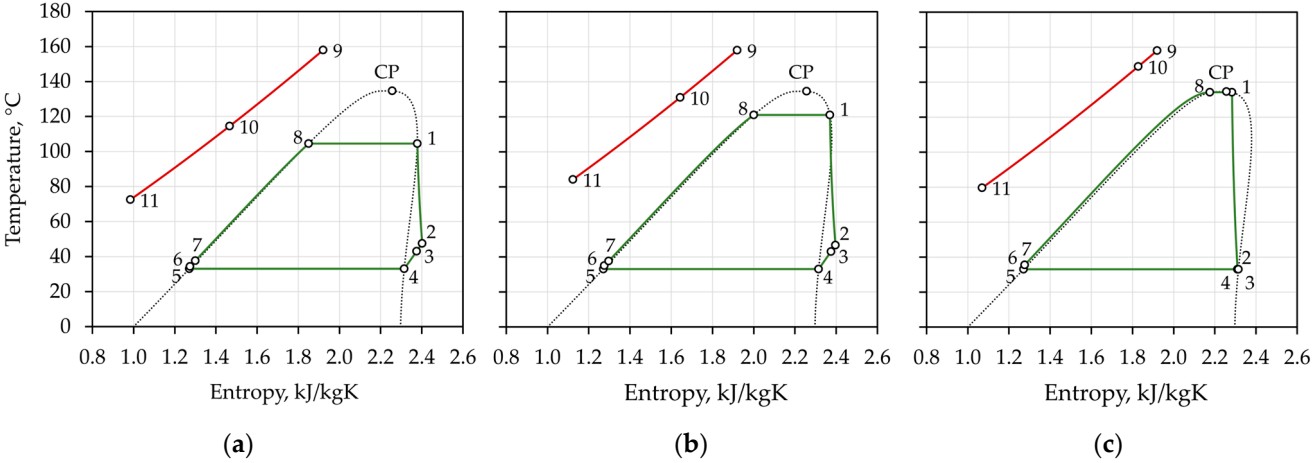

**Figure 5.** Temperature–entropy charts of subcritical saturated isobutane cycles with turbine inlet pressure of (**a**) 21.6 bar, (**b**) 28.9 bar, and (**c**) 36 bar. Results for geothermal source temperature of 158 °C and mass flow rate of 225 kg/s.

**Table 7.** Thermoeconomic performance of subcritical saturated cycles shown in Figure 5.

| Value | Turbine Inlet Pressure | | |
|---|---|---|---|
| | **21.6 bar** | **28.9 bar** | **36.0 bar** |
| Turbine inlet temperature, °C | 104.55 | 121.09 | 134.18 |
| Condenser saturation pressure, bar | 4.41 | 4.41 | 4.41 |
| Isobutane mass flow rate, kg/s | 198.97 | 169.22 | 190.03 |
| Reinjection temperature, °C | 72.50 | 84.34 | 79.87 |
| Pinch point temperature difference, °C | 10.0 | 10.0 | 10.0 |
| Evaporator heat flow rate, MW | 41.80 | 25.96 | 8.81 |
| Preheater heat flow rate, MW | 39.73 | 44.41 | 62.68 |
| Desuperheater heat flow rate, MW | 1.69 | 1.15 | 0 |
| Condenser heat flow rate, MW | 67.36 | 57.28 | 61.03 |
| Turbine gross power, MW | 10.85 | 10.52 | 11.51 |
| Gross power output, MW | 10.42 | 10.11 | 11.05 |
| Feed pump power, MW | 0.84 | 1.02 | 1.49 |
| Cooling pump power, MW | 0.60 | 0.51 | 0.55 |
| Auxiliary power, MW | 0.52 | 0.51 | 0.55 |
| Net power output, MW | 8.46 | 8.07 | 8.46 |
| Gross thermal efficiency, % | 12.78 | 14.36 | 14.78 |
| Net thermal efficiency, % | 10.37 | 11.47 | 11.32 |
| Absolute net efficiency, % | 10.08 | 9.62 | 10.08 |
| Cost of equipment, $\times 10^6$ USD$_{2022}$ | 22.61 | 20.47 | 22.13 |
| Total module costs, $\times 10^6$ USD$_{2022}$ | 30.60 | 28.44 | 31.42 |
| Initial investment, $\times 10^6$ USD$_{2022}$ | 41.91 | 38.71 | 42.48 |
| SIC, USD$_{2022}$/kW | 4955.10 | 4795.27 | 5017.36 |
| LCOE, USD$_{2022}$/MWh | 76.39 | 74.30 | 76.86 |

It should be noted that the cost analysis is particularly sensitive to the size of the surface area in the condenser because the heat transfer coefficient for film condensation is lower than those obtained in the evaporator (nucleate boiling) and in the preheater (turbulent single-phase flow). The increased equipment costs along with the limited enthalpy



difference ($h_1$–$h_2$) across the turbine expansion leads to an LCOE of 76.39 USD/MWh and a SIC of 4955.10 USD/kW, higher than in the cycle with a cost-optimum pressure of 28.9 bar.

The subcritical cycle with a turbine inlet pressure of 28.9 bar gives the worst thermal match between the geothermal fluid and isobutane. The reinjection temperature is now 84.34 °C. This results in the lowest isobutane mass flow rate (169.22 kg/s), the lowest net power (8.07 MW), and the lowest absolute net efficiency (9.62%) of the three cycles compared. However, the lower equipment cost outweighs the poorer thermal performance, and the calculated LCOE is 74.30 USD/MWh with a SIC of 4795.27 USD/kW, the lowest among the three cycles compared. The cycle with a turbine inlet pressure of 36 bar generates a net power output of 8.46 MW, thanks to a large enthalpy difference in the turbine, although the mass flow rate (190.03 kg/s) and reinjection temperature (79.87 °C) are not as large as in the cycle with a turbine inlet pressure of 21.6 bar. The LCOE of 76.86 USD/MWh and the SIC of 5017.36 USD/kW are the highest among the three cycles compared.

Isobutane cycles with dry saturated vapor in the high-pressure subcritical region should be taken with caution because isobutane is a dry working fluid. This means that the turbine expansion ends in the superheated region. The saturated vapor line exhibits a positive slope ($dT/ds > 0$) for saturation pressures below 22.9 bar and a negative slope ($dT/ds < 0$) for saturation pressure above 22.9 bar. Turbine operation with dry saturated vapor above the zero-slope point ($dT/ds = 0$ at $p_{sat}$ = 22.9 bar) could lead to excessive droplet formation and turbine blade erosion during the expansion [52,53]. This problem is particularly pronounced in the cycle with $p_1$ = 36 bar (Figure 5c) in which the expansion runs entirely through the wet vapor region, while in the cycle with 28.9 bar (Figure 5b), the expansion runs close to the dry saturated line. Two-phase turbines could operate in these conditions and offer additional power output over a single-phase system [54–56]; however, they are not yet commercially available. Erosion-related problems can be easily prevented using vapor superheating, which can also lead to improved thermoeconomic performance, as shown in the next section.

### 3.3.2. Subcritical Superheated Isobutane Cycles

The thermoeconomic performance of subcritical superheated cycles is evaluated by varying the turbine inlet pressure ($p_1$) and temperature ($T_1$), i.e., the isobutane enthalpy ($h_1$). Figure 6 shows the net power output and the LCOE for geothermal source temperatures of 190 °C. The pressure curves are plotted in the range from the dry saturated state up to 15 °C below the geothermal source temperature. The thermal optimum points are found in the superheated region, with 10–15 °C of superheat (Figure 6a). The economic analysis reveals that the minimum LCOEs are found even further in the superheated region, with superheat temperatures in the range of 160–170 °C (Figure 6b), which are 20–30 °C below the heat source temperature. The distance between the thermal and economic optimum points is explained next. For a fixed heat flow rate at the geothermal side (fixed $h_9$–$h_{11}$), more superheating increases the isobutane enthalpy differences in the preheater and evaporator ($h_1$–$h_7$) but also reduces the mass flow rate ($\dot{m}_{ORC}$), as predicted by Equations (1–2). The reduced mass flow rate causes the equipment size to decrease as well, while the higher degree of superheating increases the specific work in the turbine ($h_1$–$h_2$). An increased expansion work along with a reduced equipment cost lead to decreased LCOEs in the far superheated region.

Figure 7 compares the thermal match between the geothermal fluid and isobutane for a subcritical superheated cycle with a turbine inlet pressure of 30 bar. The temperature profiles are compared for three superheat temperatures: 127 °C, 135.8 °C, and 163 °C. The saturation temperature of isobutane at 30 bar is 123.3 °C. When the superheat temperature is $T_1$ = 135.8 °C (Figure 7b), the temperature profiles are optimally matched: the reinjection temperature is exactly $T_{11}$ = 70 °C and the PPTD is exactly $\Delta T_{pp}$ = 10 °C. The isobutane cycle achieves a maximum net power output ($\dot{W}_{net}$ = 15.05 MW). The thermal match worsens with less (Figure 7a) or more superheating (Figure 7c) because the minimum reinjection temperature or the minimum pinch point temperature difference cannot be met

anymore. In the first case (Figure 7a), the PPTD is increased to 18.5 °C in order to meet the minimum reinjection temperature of 70 °C. In the second case (Figure 7c), the reinjection temperature is increased to 100 °C to meet the minimum PPTD of 10 °C.

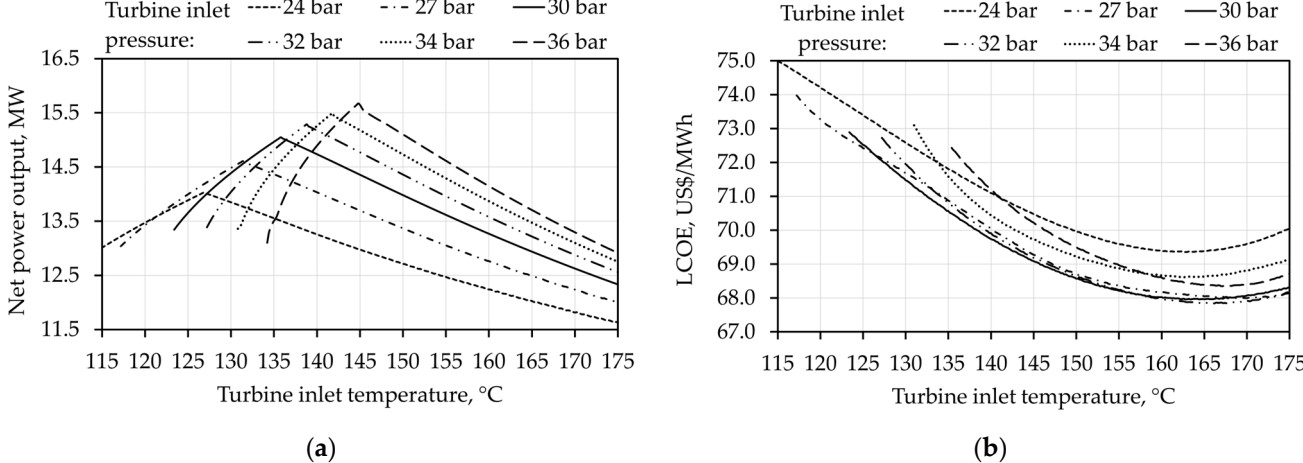

**Figure 6.** Thermoeconomic performance of subcritical superheated isobutane cycles: (**a**) net power output; (**b**) LCOE. Results for geothermal mass flow rate and temperature of 225 kg/s and 190 °C, respectively.

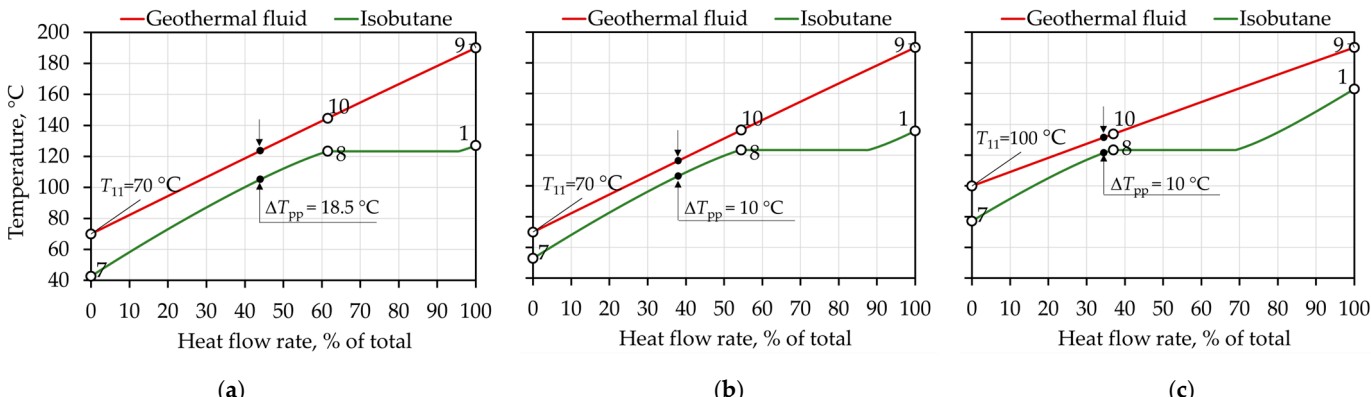

**Figure 7.** Temperature profiles in the subcritical superheated isobutane cycle with turbine inlet pressure of 30 bar and inlet temperatures of (**a**) $T_1 = 127$ °C, (**b**) $T_1 = 135.8$ °C, and (**c**) $T_1 = 163$ °C. Results for geothermal mass flow rate and temperature of 225 kg/s and 190 °C, respectively.

From Figure 7, it can be seen that the PPTD is not necessarily found at the evaporator cold end (points 8–10), but it can occur in the preheater (between points 7 and 8). This is particularly notable for high geothermal temperatures ($T_9 > 170$ °C) and high-pressure cycles ($p_1 > 30$ bar). The exact location of the PPTD is determined using an iterative subroutine (prediction-correction method) during each cycle's optimization run. The PPTD is assumed to be 10 °C in this study, whereas values between 5 and 15 °C are reported in the literature. A PPTD of less than 10 °C could increase the power output but would also lead to higher heat exchanger areas and investment costs [57].

The heat recuperator (desuperheater) affects the thermal match between geothermal brine and isobutane. In the first case, the vapor at the turbine inlet is only slightly superheated ($\Delta T_{sup} = 3.7$ °C). Therefore, less sensible heat is available for desuperheating, and liquid condensate enters the preheater with $T_7 = 42$ °C (Figure 7a). In the third case, the vapor at the turbine inlet is highly superheated ($\Delta T_{sup} = 39.7$ °C), a lot of sensible heat is available for desuperheating, and the condensate enters the preheater with $T_7 = 79$ °C (Figure 7c). It can be understood that there is just about the right degree of superheat-

ing, which leads to the maximum net power output in the subcritical superheated cycle. Figures 8 and 9 show the net power and LCOE for subcritical superheated isobutane cycles working with geothermal source temperatures of 170 °C and 150 °C.

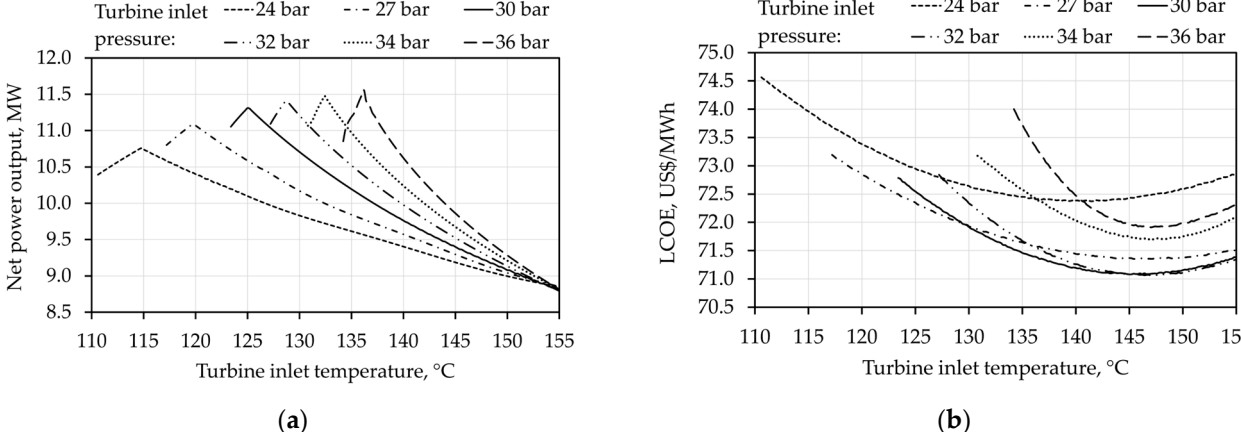

(**a**)           (**b**)

**Figure 8.** Thermoeconomic performance of subcritical superheated isobutane cycles: (**a**) net power output; (**b**) LCOE. Results for geothermal mass flow rate and temperature of 225 kg/s and 170 °C, respectively.

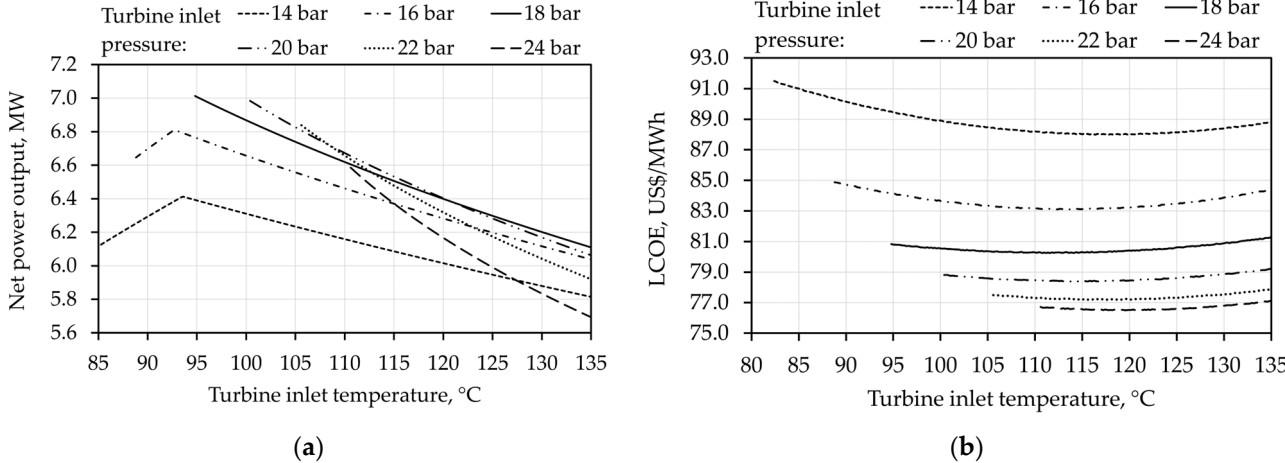

(**a**)           (**b**)

**Figure 9.** Thermoeconomic performance of subcritical superheated isobutane cycles: (**a**) net power output; (**b**) LCOE. Results for geothermal mass flow rate and temperature of 225 kg/s and 150 °C, respectively.

Figures 8 and 9 reveal that the thermal optimum points are found closer to the dry saturated line for heat source temperatures of 170 °C and 150 °C, with only 1–4 °C of superheating, while the cost optimum points are found again in the superheated region, about 20–30 °C below the heat source inlet temperature. As expected, the geothermal temperature heavily affects the net power output of the subcritical superheated cycle. The net power output is between 11.5 and 15.5 MW for a heat source temperature of 190 °C and decreases to 8.8–11.5 MW and 5.7–7.0 MW for heat source temperatures of 170 °C and 150 °C, respectively. The LCOE is less susceptible to the geothermal source temperature. The LCOE is between 68 and 75 USD/MWh for higher heat source temperatures (170 °C and 190 °C). Cycles with lower turbine inlet pressures ($p_1 \leq 18$ bar) coupled with lower geothermal temperatures ($T_9 \leq 150$ °C) result in LCOEs higher than 80 USD/MWh. Isobutane cycles operating with a heat source temperature of 130 °C (not shown here) achieve a net power output of 3.4–4.3 MW and an LCOE of 85–100 USD/MWh. Figures 6b and 8b show that medium-to-high turbine inlet pressures of 27–32 bar achieve lower LCOEs than

those of near critical pressures of 34–36 bar. Figure 9a shows that turbine inlet pressures of 16–20 bar achieve higher net power outputs than those obtained with pressures of 22–24 bar, especially in the far superheated region. All this suggests that for each heat source temperature, there may be an optimum economic cycle and likewise an optimum thermal cycle. This possibility is further explored in Section 3.3.4. Subcritical vs. Supercritical Isobutane Cycles.

The advantages of using an internal heat recuperator between the exhaust vapor and the liquid condensate are discussed below. The first of these is the higher heat flow rate absorbed by the isobutane cycle, which increases its mass flow rate and the cycle net power output. The condenser size is reduced since less waste heat is discharged by the cooling towers. The preheater size is reduced as well. The LCOE decreases as a result of the reduced heat transfer surface areas, although a part is offset by the newly added heat recuperator surfaces. The advantages of the heat recuperator become more pronounced as the turbine inlet state moves further in the superheated region. For a geothermal temperature of $T_9 = 190$ °C, the superheated recuperated isobutane cycle with $p_1 = 30$ bar and $T_1 = 135.8$ °C achieves a maximum net power output of 15.05 MW with an LCOE of 70.86 USD/MWh and a SIC of 4470.3 USD/kW. The same non-recuperated cycle achieves the thermal optimum point further into the superheated region ($p_1 = 30$ bar, $T_1 = 153.9$ °C). However, the thermoeconomic performance deteriorates: the net output is 13.51 MW ($-10.2$%) and the LCOE is 72.33 USD/MWh ($+2.1$%), while the SIC is reduced slightly to 4397.9 USD/kW ($-1.6$%). This suggests that total initial costs are increased since the cost of the internal heat recuperator module outweighs the added power capacity. However, in the long term, the additional revenues from the increased electricity generation will pay off the investment in the heat recuperator.

The positive effects from the internal heat recuperator decrease as the turbine inlet state moves closer to the dry saturated curve. For example, a dry saturated isobutane cycle ($p_1 = 30$ bar, $T_1 = 123.3$ °C) with heat recuperation generates 13.32 MW at an LCOE of 73.21 USD/MWh and at a SIC of 4445.3 USD/kW, while the same non-recuperated cycle generates 13.13 MW with an LCOE of 73.28 USD/MWh and a SIC of 4422.4 USD/kW.

### 3.3.3. Supercritical Cycles

Figures 10 and 11 show the thermoeconomic performance of supercritical isobutane cycles in terms of the net power output and LCOE for different pressures ($p_1$) and temperatures ($T_1$) at the turbine inlet and for geothermal source temperatures of 190 °C and 170 °C. The critical point of isobutane is defined by a critical temperature of $T_{cr} = 134.67$ °C and a critical pressure of $p_{cr} = 36.29$ bar. Supercritical isobutane cycles exhibit maximum net power values similarly to the subcritical superheated cycles at higher heat source temperatures (Figures 6 and 8). For supercritical cycles, thermal optimum points are found slightly on right of the critical point in the *T-s* chart ($s_1 \gtrsim s_{cr}$). The economic analysis, on the other hand, reveals that the lowest LCOEs are found in the far superheated region, similarly to the subcritical superheated cycles. Supercritical pressures slightly above the critical pressure ($p_1 = 37$–41 bar) achieve lower LCOEs than higher supercritical pressures ($p_1 = 43$–47 bar), because the latter are penalized with increased equipment costs (higher pressure factor $f_P$ in Equation (23)) and a larger feed pump.

High-pressure subcritical cycles as well as supercritical cycles use slightly superheated vapor at the turbine inlet when optimized for maximum net power. Figure 12 shows the *T-s* charts of three isobutane cycles optimized for maximum net power output. The first cycle uses subcritical isobutane with a turbine inlet pressure and temperature of $p_1 = 30$ bar and $T_1 = 125$ °C. This cycle achieves a net power output of 11.31 MW with an LCOE of 72.67 USD/MWh. The second cycle uses transcritical isobutane with a turbine inlet pressure and temperature of $p_1 = 37$ bar and $T_1 = 137.8$ °C. It achieves a net power of 11.51 MW and an LCOE of 73.71 USD/MWh. The third cycle uses supercritical isobutane with $p_1 = 41$ bar and $T_1 = 144.3$ °C and achieves a net power of 11.48 MW with an electricity cost of 74.76 USD$_{2022}$/MWh.

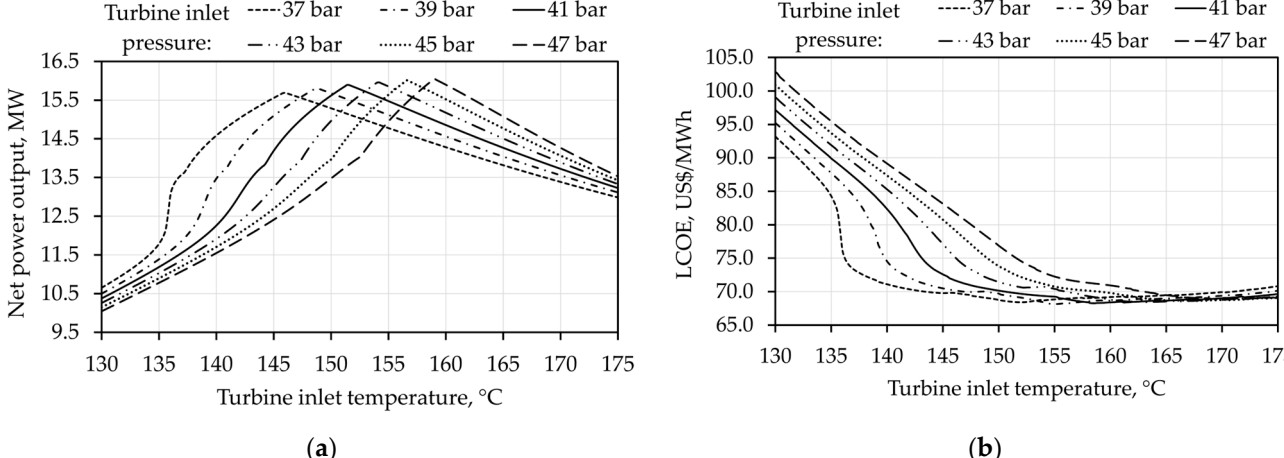

**Figure 10.** Thermoeconomic performance of supercritical isobutane cycles: (**a**) net power output and (**b**) LCOE. Results for geothermal mass flow rate and temperature of 225 kg/s and 190 °C, respectively.

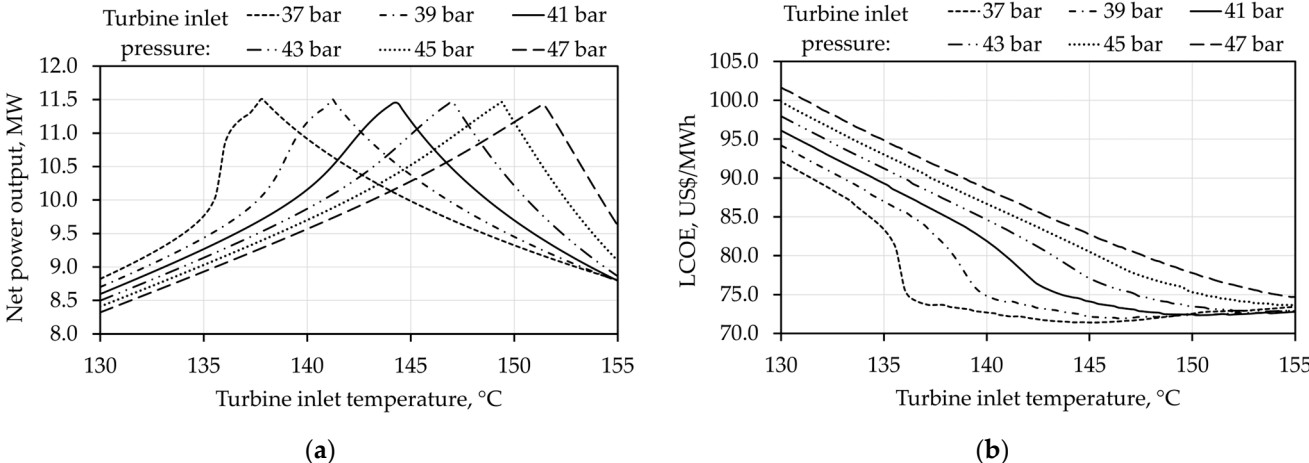

**Figure 11.** Thermoeconomic performance of supercritical isobutane cycles: (**a**) net power output and (**b**) LCOE. Results for geothermal mass flow rate and temperature of 225 kg/s and 170 °C, respectively.

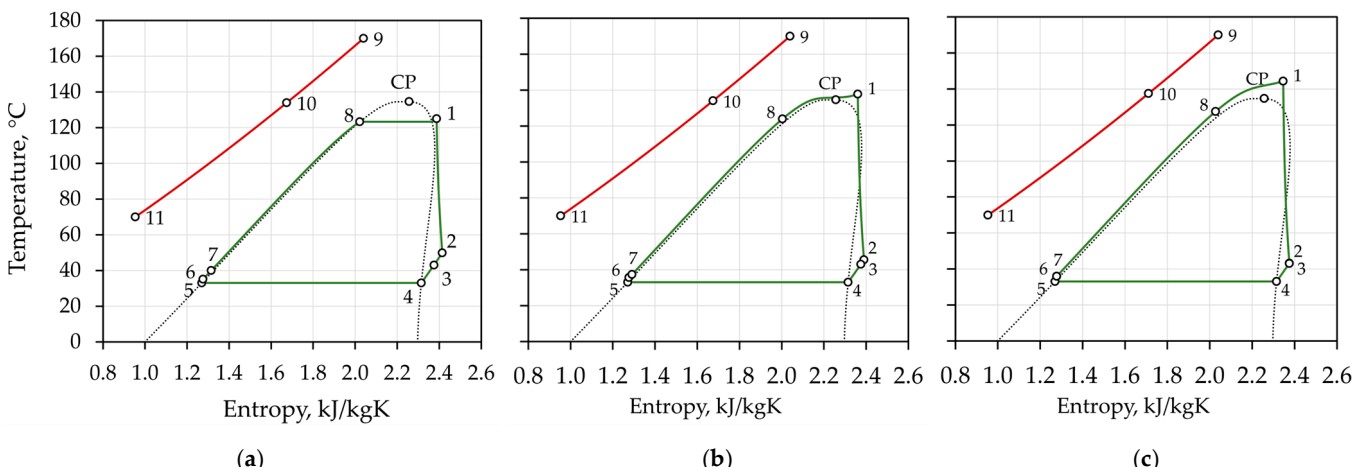

**Figure 12.** Temperature–entropy charts of three isobutane cycles optimized for maximum net power output: (**a**) $p_1 = 30$ bar, $T_1 = 125$ °C; (**b**) $p_1 = 37$ bar, $T_1 = 137.8$ °C; and (**c**) $p_1 = 41$ bar, $T_1 = 144.3$ °C. Results for geothermal mass flow rate and temperature of 225 kg/s and 170 °C, respectively.

Compared to subcritical cycles, supercritical isobutane cycles benefit from improved thermal performance when cycle optimization is performed for the maximum power output. However, the *T-s* charts reveal that these cycles expand partially through the wet vapor region (Figure 12b,c), and additional superheat is necessary to prevent turbine blade erosion. The subcritical and supercritical cycles are further compared, imposing a minimum superheat temperature of 5 °C from the dry saturated curve along the entire expansion curve. Figure 13 shows the updated *T-s* charts of the three isobutane cycles with a minimum superheat temperature of $\Delta T_{\text{sup}} = 5$ °C. The subcritical cycle achieves a net power of 10.90 MW with an LCOE of 72.10 USD/MWh. The transcritical cycle achieves a net power output of 10.59 MW with an LCOE of 72.17 USD/MWh, while the supercritical cycle achieves 10.07 MW with 72.64 USD/MWh. The last two cycles are particularly affected by the additional superheating because they depart from the thermal optimum point. The net power outputs are reduced by 8.7% ($p_1 = 37$ bar) and 12.3% ($p_1 = 41$ bar). The subcritical cycle is the least affected by the additional superheat temperature: the net power reduction is only 3.6%, and this cycle achieves the best overall thermoeconomic performance among the three compared cycles.

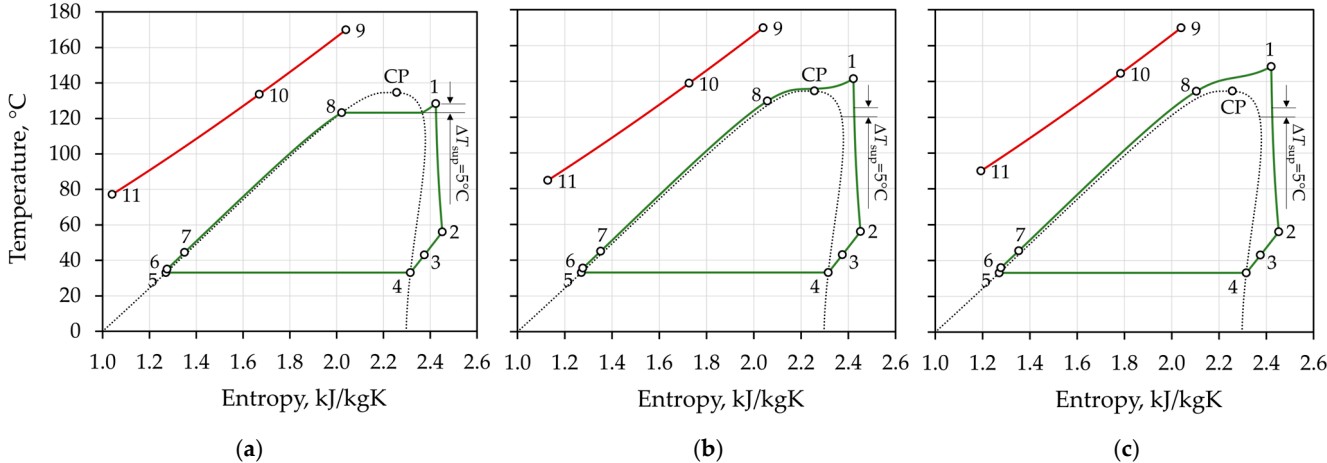

**Figure 13.** Temperature–entropy charts of three isobutane cycles respecting a minimum superheat temperature of $\Delta T_{\text{sup}} = 5$ °C along the turbine expansion: (**a**) $p_1 = 30$ bar, $T_1 = 128.3$ °C; (**b**) $p_1 = 37$ bar, $T_1 = 141.5$ °C; and (**c**) $p_1 = 41$ bar, $T_1 = 148.3$ °C.

The departure from the thermal optimum point causes a worsened thermal match between the geothermal fluid and working fluid. The additional superheating causes the reinjection temperatures to increase above the minimum allowed value (Figure 13): $T_{11} = 77.2$ °C for $p_1 = 30$ bar, $T_{11} = 84.6$ °C for $p_1 = 37$ bar, and $T_{11} = 90.1$ °C for $p_1 = 41$ bar. All the reinjection temperatures are $T_{11} = 70$ °C in the thermal optimum cycles, as seen in Figure 12.

### 3.3.4. Subcritical vs. Supercritical Isobutane Cycles

In this section, the thermoeconomic performance of subcritical and supercritical cycles is compared for geothermal source temperatures of $T_{\text{geo}} = 100$–200 °C and for a mass flow rate of $\dot{m}_{\text{geo}} = 225$ kg/s. Figure 14 shows the turbine inlet pressure ($p_1$) and the temperature ($T_1$), while Figure 15 shows the net power output and the economic indicators (LCOE and SIC) for optimum isobutane cycles. Subcritical and supercritical isobutane cycles are optimized for maximum power generation and have a minimum superheat of $\Delta T_{\text{sup}} = 5$ °C along the turbine expansion. For geothermal temperatures of $T_{\text{geo}} < 179$ °C, the maximum net power is obtained with the subcritical isobutane cycles. For geothermal temperatures $T_{\text{geo}} \geq 179$ °C, the thermal optimum is obtained with the supercritical isobutane. For $T_{\text{geo}} = 179$ °C, the optimum turbine inlet condition ($p_1 = 36.5$ bar and $T_1 = 139.9$ °C) is just above the critical point, while for a $T_{\text{geo}} = 150$ °C, the optimum turbine inlet pressure

and temperature are $p_1 = 18.1$ bar and $T_1 = 100$ °C. The distance between the turbine inlet temperature curve and the saturation temperature curve corresponds to $\Delta T_{sup} = 5$ °C.

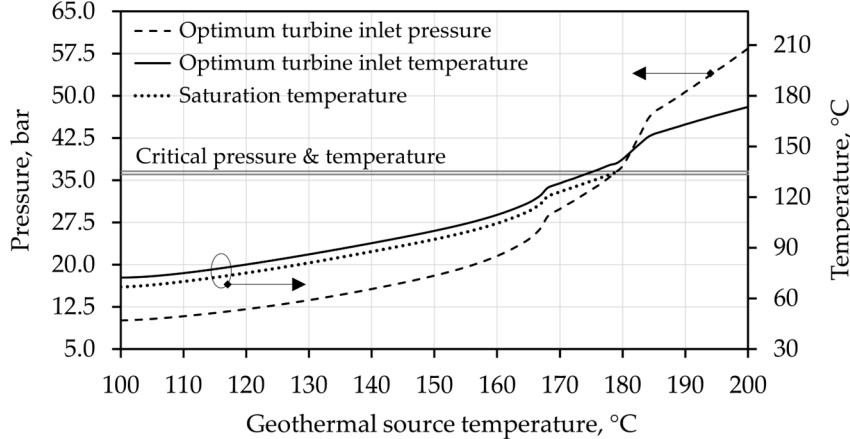

**Figure 14.** Optimum turbine inlet pressure and temperature for maximum power generation.

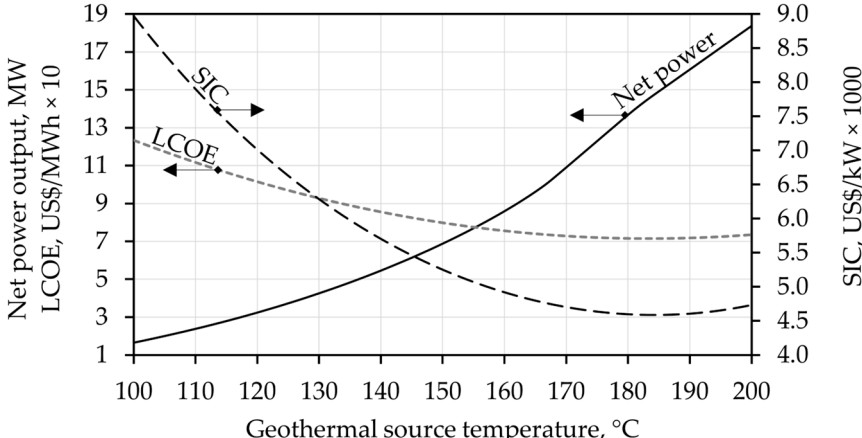

**Figure 15.** Net power output, LCOE, and SIC of thermal optimum cycles.

The economic performance is strongly affected by the temperature of the geothermal source. When the source temperature is $T_{geo} \leq 130$ °C, the average estimated LCOE is over 90 USD/MWh, while the SIC is over 6300 USD/kW. At geothermal temperatures of $T_{geo} \geq 150$ °C, the isobutane cycles achieve an LCOE and SIC below 80 USD/MWh and 5250 USD/kW, respectively.

## 4. Conclusions

The following highlights arise from the thermoeconomic analysis of subcritical and supercritical isobutane cycles for geothermal power generation:

- The thermal optimum point (maximum net power output) is different from the economic optimum point (minimum LCOE);
- For geothermal source temperatures of 100–158 °C, the thermal optimum points are found on the dry saturated curve of subcritical isobutane cycles;
- For geothermal source temperatures of 158–179 °C, the thermal optimum points are obtained with slightly superheated subcritical cycles' vapor ($\Delta T_{sup} < 5$ °C);
- Only for higher geothermal source temperatures (above 179 °C) do supercritical isobutane cycles achieve a better thermal performance (higher net power output) than subcritical cycles while the economic performance (LCOE and SIC) is comparable;

- Binary cycles with near-critical pressures at the turbine inlet expand through the wet vapor region. While vapor superheating eliminates the risk of droplet formation and turbine blade erosion, it also causes departure from the thermal optimum point;
- For subcritical superheated isobutane cycles, the economic optimum point is found further in the superheated region relative to the thermal optimum point;
- Internal heat recuperation offers an improved thermoeconomic performance over non-recuperated cycles: the net power increases and the LCOE decreases, although installation costs (SIC) increase because of the costs of the additional heat exchanger;
- The estimated LCOE and SIC of isobutane binary cycles depend on the geothermal source temperature ($T_{geo}$) and mass flow rate ($\dot{m}_{geo}$), i.e., the power plant size;
- For a geothermal source with $\dot{m}_{geo} = 225$ kg/s and $120 < T_{geo} < 150\ ^{\circ}$C, the estimated costs are 100 > LCOE > 80 USD/MWh and 7000 > SIC > 5250 USD/kW, while for medium–high geothermal temperatures of $150 < T_{geo} < 200\ ^{\circ}$C, the estimated costs are 80 > LCOE > 70 USD/MWh and 5250 > SIC > 4600 USD/kW.

The present study developed a novel and detailed thermoeconomic model for the analysis of subcritical and supercritical isobutane cycles in binary geothermal power plants. The model was validated against publicly available data of real geothermal power plants. The model proved capable of comprehensively evaluating the performance of binary cycles over a wide range of geothermal source temperatures, cycle operating conditions, and isobutane states. Further, cycle optimization can be carried out by abiding by the minimum superheat temperature necessary to avoid turbine blade erosions, which has been regularly neglected in the literature. The obtained results in this study are limited to single-stage cycles generating geothermal electricity from heat source temperatures in the range between 100 and 200 °C. The model is primarily developed for isobutane as the selected working fluid, but it could be used on other pure and mixed working fluids after minor adjustments. The obtained results can be considered as average estimates while the conclusions should be seen as general guidelines for engineers developing binary geothermal power plants. For a more detailed analysis of the geothermal project feasibility, case-specific features should be included as well, such as the geothermal site properties, the power plant location and the macroeconomic environment, the labor and maintenance costs, electricity retail prices, and feed-in tariff (FIT) policies. Future research should aim to improve the thermoeconomic approach with higher levels of flexibility and details. A second line of research could be extending the thermoeconomic analysis onto advanced binary cycle configurations, different working fluids and mixtures, or other factors influencing the thermoeconomic performance of binary cycle technologies.

**Author Contributions:** Conceptualization, A.A.B. and P.B.; methodology, A.A.B. and P.B.; software, P.B.; validation, A.A.B. and P.B.; formal analysis, A.A.B. and P.B.; investigation, A.A.B. and P.B.; resources, A.A.B.; data curation, P.B.; writing—original draft preparation, A.A.B. and P.B.; writing—review and editing, A.A.B. and P.B.; visualization, P.B.; supervision, A.A.B.; project administration, A.A.B.; funding acquisition, A.A.B. and P.B. All authors have read and agreed to the published version of the manuscript.

**Funding:** This research received no external funding.

**Institutional Review Board Statement:** Not applicable.

**Informed Consent Statement:** Not applicable.

**Data Availability Statement:** Not applicable.

**Conflicts of Interest:** The authors declare no conflict of interest.

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
