# Peer review of "Thermoeconomic Analysis of Subcritical and Supercritical Isobutane Cycles for Geothermal Power Generation"

_sustainability, doi:10.3390/su15118624_

Round 1
Reviewer 1 Report
In this manuscript, the authors developed a thermoeconomic approach to evaluate the performance of subcritical and supercritical cycles in the geothermal temperature range of 100~200 °C.
The thermoeconomic model could evaluate the performance of binary cycles over a wide range of operating situations. Moreover, the model was applied to determine and compare thermal and economic optimum conditions, and analyze superheated and recuperated isobutane cycles across the subcritical as well as supercritical pressure regions. The obtained results can be treated as average estimates while the conclusions as general guidelines for researchers developing binary geothermal power plants.
The literature is well-described, making readers understand a brief studying history of this field.
In summary, the manuscript can be published after minor revision.
Some minor suggestions are listed below.
-----------------------------------------------
-----------------------------------------------
[[Minor suggestions]]
The suggestions are structured as shown below.
[Suggested point][Position]
Descriptions.
1. [Abbreviation][Line 56]
The abbreviation (ORC) should be stated with the full name before using it.
2. [Notation][Line 177 and 178]
Please modify the dot position of the terms “m_ORC” and m_GTF.”
2. [Notation][Line 179]
Please check the term “eta_PR”
Is it “eta_PH”?
Reviewer 2 Report
This paper puts forward a comprehensive model for the thermoeconomic performance of geothermal power generation plants, specifically using subcritical and supercritical isobutane as working fluid. Optimum operating conditions for targeting either power output or costs are obtained. The paper is well organized and the results are valid. The scope fits in the journal's topic. The paper is recommended for publication after minor revision, addressing comments below.
1. ORC in line 56 is not defined.
2. "and 48 kg/s" in line 60 should be "of 48 kg/s".
3. Reference is recommended for the assumption of using 10% in line 179.
4. Reference is recommended for the DCF model in section 2.4.
5. In figure 2, it would be interesting if any real data points can be shown on the plot.
6. In the last conclusion point, the values do not agree with the direction of "<" sign.
Language is overall good. There are minor issues. Further proofreading is recommended for language improvement.
Reviewer 3 Report
1- The abstract needs to be revised. Instead of a demonstration of some numbers please point out the results and its innovation compared to prior works.
2- English has many grammatical and typical mistakes. For this journal, we need excellent writing.
3- The history of the research needs to be improved and some new research areas including artificial intelligence need to be mentioned. The following works are highly recommended to add:
Modern Adaptive Fuzzy Control Systems
Neural Networks and Learning Algorithms in MATLAB
4- Some of your equations need references, proof, or better clarification. If they are from a theorem need to give the main reference at the beginning of your section.
5- How tables 2 and 3 are given? please clarify.
6- Using equation 22 and 23 are questionable here. please clarify it better.
7-"3.1. Validation of the thermodynamic model" needs to be improved.
8- Please point out the limitations of your work better and bring it in your text.
9-What is your future suggestion for this work?
10-Please in the conclusion, mention the comparison and your work novelty better.
11 - Please clarify figures 12 and 13 in your text better.
English has many grammatical and typical mistakes. For this journal, we need excellent writing.
Round 2
Reviewer 3 Report
Using artificial intelligence for your future works is recommended.